# Small molecule inhibition of apicomplexan FtsH1 disrupts plastid biogenesis in human pathogens

Katherine Amberg-Johnson[1,2], Sanjay B Hari[3], Suresh M Ganesan[4], Hernan A Lorenzi[5], Robert T Sauer[3], Jacquin C Niles[4], Ellen Yeh[1,2,6]*

[1]Department of Biochemistry, Stanford Medical School, Stanford, United States; [2]Microbiology and Immunology, Stanford Medical School, Stanford, United States; [3]Department of Biology, Massachusetts Institute of Technology, Cambridge, United States; [4]Department of Biological Engineering, Massachusetts Institute of Technology, Cambridge, United States; [5]Department of Infectious Disease, The J. Craig Venter Institute, Maryland, United States; [6]Department of Pathology, Stanford Medical School, Stanford, United States

**Abstract** The malaria parasite *Plasmodium falciparum* and related apicomplexan pathogens contain an essential plastid organelle, the apicoplast, which is a key anti-parasitic target. Derived from secondary endosymbiosis, the apicoplast depends on novel, but largely cryptic, mechanisms for protein/lipid import and organelle inheritance during parasite replication. These critical biogenesis pathways present untapped opportunities to discover new parasite-specific drug targets. We used an innovative screen to identify actinonin as having a novel mechanism-of-action inhibiting apicoplast biogenesis. Resistant mutation, chemical-genetic interaction, and biochemical inhibition demonstrate that the unexpected target of actinonin in *P. falciparum* and *Toxoplasma gondii* is FtsH1, a homolog of a bacterial membrane AAA+ metalloprotease. *Pf*FtsH1 is the first novel factor required for apicoplast biogenesis identified in a phenotypic screen. Our findings demonstrate that FtsH1 is a novel and, importantly, druggable antimalarial target. Development of FtsH1 inhibitors will have significant advantages with improved drug kinetics and multistage efficacy against multiple human parasites.
DOI: https://doi.org/10.7554/eLife.29865.001

*For correspondence:
ellenyeh@stanford.edu

## Introduction

There is an urgent need for antimalarials with novel mechanisms-of-action to combat resistance to frontline drugs. The apicoplast is a plastid organelle uniquely found in *Plasmodium* parasites that cause malaria and related apicomplexan pathogens (*McFadden et al., 1996*; *Köhler et al., 1997*). Though non-photosynthetic, the apicoplast is required for the biosynthesis of essential metabolites in the organisms that retained it (*Jomaa et al., 1999*; *Mazumdar et al., 2006*; *Vaughan et al., 2009*; *Yu et al., 2008*; *Nair et al., 2011*; *Yeh and DeRisi, 2011*; *Ke et al., 2014*). Acquired by a secondary eukaryote-eukaryote endosymbiosis, it utilizes bacterial and organellar pathways distinct from those of mammalian host cells. Given its unique and essential biology, the apicoplast is an ideal source of parasite-specific drug targets that minimize host off-target toxicity.

Despite its biomedical potential, development of broadly effective antimalarials targeting the apicoplast has met with significant roadblocks. The first class of apicoplast inhibitors to be identified was drugs like doxycycline, clindamycin, and chloramphenicol that block prokaryotic protein synthesis (*Fichera and Roos, 1997*; *Dahl et al., 2006*). The antimalarial effects of these clinically-approved antibiotics were noticed well before they were shown to inhibit the bacterial-like ribosome in the

apicoplast (*Geary and Jensen, 1983*). Unfortunately translation inhibitors cause a characteristic 'delayed death' in vitro in which parasite growth inhibition occurs in the second replication cycle after drug treatment. The delayed effect is related to their mechanism-of-action and is also seen with DNA replication and transcription inhibitors that block apicoplast gene expression. Delayed death manifests as a slow onset-of-action of translation inhibitors that limits their antimalarial efficacy and clinical use.

The next druggable targets in the apicoplast were several prokaryotic metabolic pathways. Fosmidomycin, an inhibitor of MEP isoprenoid precursor biosynthesis in the apicoplast, causes parasite growth inhibition in a single replication cycle in vitro and rapid parasite clearance in human clinical trials (*Jomaa et al., 1999*; *Oyakhirome et al., 2007*; *Lanaspa et al., 2012*; *Guggisberg et al., 2016*). Unfortunately initial parasite clearance is followed by recrudescent infections in 50% of patients. These treatment failures have brought into question the clinical utility of fosmidomycin and other MEP inhibitors. Yet despite the expectation that the apicoplast would serve many essential functions, we showed that only isoprenoid precursor biosynthesis is required during the symptomatic blood stage of *Plasmodium* (*Yeh and DeRisi, 2011*). The apicoplast's limited function in blood-stage *Plasmodium* precludes opportunities to target alternative metabolic pathways.

The pipeline for essential and druggable apicoplast targets has run dry. New approaches are needed to identify drug targets that avoid the known liabilities of targeting apicoplast gene expression and metabolism. Until now, identification of drug targets in the apicoplast has been based on conserved bacterial and primary chloroplast pathways whose functions can be inferred from model organisms. However, many essential pathways for protein/lipid import into the apicoplast and organelle inheritance during parasite replication will be unique to secondary plastids and evolutionarily divergent from model eukaryotic biology (*Waller et al., 1998*; *Vaishnava and Striepen, 2006*). Drug targets in these organelle biogenesis pathways are likely to be (1) essential to parasite survival, even in the context of the apicoplast's drastically reduced metabolic function, (2) required for all proliferative stages of the parasite life cycle, and (3) conserved across apicomplexan parasites. Thus apicoplast biogenesis represents a promising but unexplored frontier in antimalarial drug discovery.

While forward genetic screens have been extremely powerful in uncovering novel cellular pathways in model organisms, they are not currently tractable in *Plasmodium*. To circumvent these limitations, we took a chemical-genetic approach from phenotypic screening to target identification. We first identified the natural product actinonin as a novel inhibitor of apicoplast biogenesis. We then uncovered the essential role of the membrane metalloprotease FtsH1 in organelle biogenesis in apicomplexan pathogens. FtsH1 is a druggable target in apicoplast biogenesis that offers advantages over the mechanisms-of-action of existing apicoplast inhibitors and has a ready hit compound to pursue drug development.

## Results

### Identification of a novel apicoplast biogenesis inhibitor

We screened >400 growth-inhibitory antimalarial compounds with unknown mechanisms-of-action to identify novel inhibitors of apicoplast biogenesis (*Figure 1—source data 1*) (*Spangenberg et al., 2013*; *Wiesner et al., 2001*; *Dahl and Rosenthal, 2007a*; *Wu et al., 2015*; *Gisselberg et al., 2017*). Our selection criteria aimed to identify compounds that (1) caused parasite growth inhibition (essential target) in a single replication cycle (avoiding delayed death), (2) were specific in their disruption of the apicoplast, and (3) resulted in loss of the apicoplast during parasite replication. This unique inhibition phenotype distinguished selected compounds from known apicoplast gene expression or metabolism inhibitors, ensuring discovery of a novel class of inhibitors.

A single natural product antibiotic, actinonin, matched our criteria (*Figure 1a*) (*Gordon et al., 1962*). Consistent with its reported antimalarial activity, actinonin caused *P. falciparum* growth inhibition in a single replication cycle ($EC_{50}$ = 3.2 μM; 95% CI 2.5–4.1) (*Wiesner et al., 2001*; *Goodman and McFadden, 2014*). Because isoprenoid precursor biosynthesis is the only essential function of the apicoplast in blood-stage *P. falciparum*, any disruption of the apicoplast, including complete loss of the organelle, can be rescued by addition of the isoprenoid precursor, isopentenyl pyrophosphate (IPP), in the growth media (*Yeh and DeRisi, 2011*). Indeed actinonin's $EC_{50}$ shifted nearly 20-fold in the presence of IPP to 61 μM (95% CI 50–75). Although the translation inhibitor

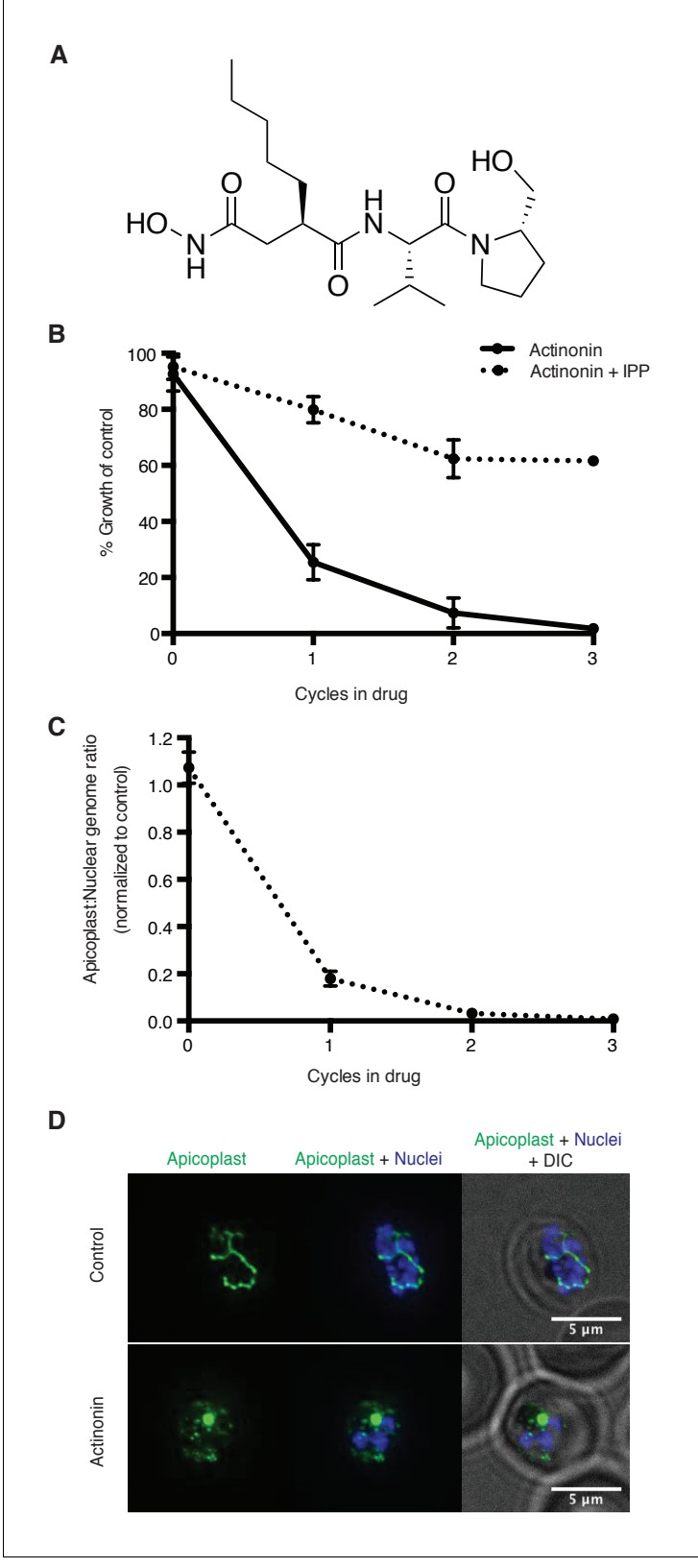

**Figure 1.** Actinonin inhibits apicoplast biogenesis in *P. falciparum*. (a) Structure of actinonin. (b) Time course of parasite growth during actinonin treatment with or without IPP, normalized to control cultures with or without IPP as appropriate. Error bars represent the SEM of two biological replicates. (c) Time course of the apicoplast:nuclear

*Figure 1 continued on next page*

*Figure 1 continued*

genome ratio measured by quantitative PCR (qPCR) using primers for the apicoplast and nuclear genomes during treatment with actinonin and IPP. Genome ratios were normalized to control parasites grown with IPP only. Error bars as in b. (**d**) Representative images of the apicoplast of IPP-rescued control and actinonin treated parasites 24 hr after treatment during the schizont stage. The apicoplast is visualized using the *P. falciparum* reporter strain D10 ACP-GFP in which GFP is targeted to the apicoplast and the nucleus is stained with Hoescht 33342. During *Plasmodium* replication, the apicoplast starts as a single small spherical organelle (ring stage) which branches and divides into multiple apicoplasts (schizont stage). A punctate apicoplast that does not branch indicates a defect in apicoplast biogenesis.

DOI: https://doi.org/10.7554/eLife.29865.002

The following source data and figure supplements are available for figure 1:

**Source data 1.** Antimalarial activity of inhibitors screened in this study.
DOI: https://doi.org/10.7554/eLife.29865.006
**Source data 2.** Numerical data for *Figure 1* and *Figure 1—figure supplement 1* .
DOI: https://doi.org/10.7554/eLife.29865.007
**Figure supplement 1.** Actinonin specifically inhibits the apicoplast of *P. falciparum* leading to parasite death after a single replication cycle.
DOI: https://doi.org/10.7554/eLife.29865.003
**Figure supplement 2.** Actinonin has a distinct inhibition phenotype compared to inhibitors of apicoplast metabolism and translation.
DOI: https://doi.org/10.7554/eLife.29865.004
**Figure supplement 3.** Actinonin is unlikely to inhibit the peptide deformylase of *P. falciparum*.
DOI: https://doi.org/10.7554/eLife.29865.005

chloramphenicol also caused growth inhibition rescued by IPP, parasite growth inhibition and resulting apicoplast biogenesis defects were delayed by one replication cycle following drug treatment (*Figure 1—figure supplement 1b*; *Figure 1—figure supplement 2a,c*). Compared to chloramphenicol, actinonin clearly had more rapid antimalarial activity. These results demonstrate that actinonin specifically inhibits the apicoplast without a delayed death phenotype (*Figure 1b*; *Figure 1—figure supplement 1a*; *Figure 1—source data 1*).

Previously actinonin was observed to cause a block in the morphologic development of the apicoplast during parasite replication (*Goodman and McFadden, 2014*). However, these defects were observed without IPP growth rescue when parasite replication was also inhibited. Absent IPP rescue, the MEP inhibitor fosmidomycin also showed a block in apicoplast development. Based on the observed similarities in their inhibition phenotype, the authors concluded that actinonin, like fosmidomycin, inhibited apicoplast metabolism. We sought to clarify these earlier observations using IPP rescue to distinguish between specific apicoplast defects caused by the inhibitor and nonspecific defects observed in a non-replicating parasite. Using quantitative PCR (qPCR) to detect the apicoplast genome and microscopy to localize an apicoplast-targeted GFP, we show that actinonin-treated *P. falciparum*, rescued for growth by IPP, no longer replicated their apicoplast genome and produced daughter parasites lacking apicoplasts, consistent with a defect in apicoplast biogenesis (*Figure 1c–d*; *Figure 1—figure supplement 2a,e*) (*Yeh and DeRisi, 2011*). In contrast, both the copy number of the apicoplast genome and apicoplast development were unaffected by fosmidomycin treatment under IPP rescue, indicating that the previously reported apicoplast biogenesis defects were nonspecific (*Figure 1—figure supplement 1b*; *Figure 1—figure supplement 2a,d*). Taken together, actinonin's inhibition phenotype distinguishes it from known inhibitors that disrupt apicoplast gene expression and metabolism and indicates that it has a novel mechanism-of-action in organelle biogenesis (*Figure 1—figure supplement 1b*; *Figure 1—figure supplement 2*).

## Actinonin is unlikely to inhibit the peptide deformylase in *Plasmodium* parasites

To identify actinonin's molecular target, we first took a candidate-based approach. In bacteria and mammalian mitochondria, actinonin potently inhibits a peptide deformylase (PDF) which catalyzes the co-translational removal of the formyl group from nascent chains (*Chen et al., 2000*). The apicoplast translation machinery is prokaryotic in origin and therefore contains a PDF. Previously actinonin

was shown to inhibit the enzymatic activity of purified *P. falciparum* PDF with a reported $IC_{50}$ = 2.5 μM (*Bracchi-Ricard et al., 2001*). The potency of this inhibition is $10^{2-3}$ fold less than those reported for *E. coli* or human PDF (*Chen et al., 2000*; *Lee et al., 2004*; *Kumar et al., 2002*). Given the role of *Pf*PDF in apicoplast translation, its inhibition is expected to give a delayed death phenotype as observed with other translation inhibitors, whereas we clearly observed growth inhibition in a single replication cycle (*Figure 1*). Puzzled by this inconsistency, we used both genetic and chemical approaches to ascertain whether the *Pf*PDF is the target of actinonin.

In bacteria, resistance to actinonin is conferred by overexpression of PDF (*Margolis et al., 2000*). We constructed a *P. falciparum* strain containing a second copy of *Pf*PDF-myc in which expression was regulated by tetR-DOZI-binding aptamer sequences (*Ganesan et al., 2016*). As expected, *Pf*PDF-myc expression was observed in the presence of anhydrotetracycline, which disrupts the tetR-DOZI repressor-aptamer interaction (*Figure 1—figure supplement 3a*). We assessed the import of overexpressed *Pf*PDF-myc into the apicoplast by determining its apicoplast-dependent protein cleavage. Lumenal apicoplast proteins, including *Pf*PDF, contain an *N*-terminal transit peptide sequence that is required for trafficking to the apicoplast and is cleaved upon import (*Waller et al., 1998*). In normal parasites, *Pf*PDF-myc was detected as a majority processed and minority unprocessed protein (*Figure 1—figure supplement 3a–b*). In IPP-rescued parasites lacking their apicoplast, *Pf*PDF-myc accumulates as only the unprocessed protein (*Figure 1—figure supplement 3b*) (*Yeh and DeRisi, 2011*). These results indicate that *Pf*PDF-myc is properly trafficked. Contrary to what is seen in bacteria, *Pf*PDF-myc overexpression did not confer actinonin resistance indicating that *Pf*PDF may not be the target of actinonin (*Figure 1—figure supplement 3c*). Because the lack of actinonin resistance may also have been due to insufficient overexpression or nonfunctional *Pf*PDF-myc, we attempted to generate a strain with *pdf* expression regulated at the endogenous loci but were unable obtain viable parasites in two separate transfections performed in parallel with successful positive controls.

We also assessed whether the actinonin target functions downstream of the ribosome, as is expected for *Pf*PDF. In mitochondria, translation inhibitors suppress the effects of actinonin since a block in translation supercedes a block in PDF activity (*Richter et al., 2013*). Similarly, we expected that if PDF was the target of actinonin leading to growth inhibition in a single replication cycle, then co-treatment with chloramphenicol would result in a delayed death phenotype, whereby translation inhibition would mask the effects of PDF inhibition. Surprisingly, chloramphenicol did not suppress the effects of actinonin (*Figure 1—figure supplement 3d*). Taken together, our results indicate that (1) actinonin does not cause delayed death as would be expected for *Pf*PDF inhibition, (2) overexpression of *Pf*PDF did not confer actinonin resistance, and (3) the effect of actinonin was not suppressed by inhibition of the ribosome upstream of *Pf*PDF. These results are inconsistent with *Pf*PDF being the target of actinonin.

## Actinonin-resistant *T. gondii* contain a mutation in *Tg*FtsH1

Because the target of actinonin in *P. falciparum* did not appear to be conserved in model organisms, we turned to an unbiased approach. Our collaborators and we independently attempted to isolate actinonin-resistant *P. falciparum* but failed using multiple selection methods, including chemical mutagenesis, that successfully selected resistance against other compounds (Prof. David Fidock, personal communication, July 2017) (*Wu et al., 2015*; *Gisselberg et al., 2017*). Therefore, we turned to *Toxoplasma gondii,* a related apicomplexan parasite, which contains an apicoplast of the same evolutionary origin, because it is easier to grow to large numbers and to genetically modify.

To determine whether actinonin disrupts apicoplast biogenesis in *T. gondii*, we characterized its inhibition phenotype. Curiously, *T. gondii* also shows a delayed death phenotype with apicoplast translation inhibition, whereby drug treatment in the first lytic cycle results in parasite growth inhibition in the second cycle (*Fichera and Roos, 1997*). However, the mechanism underlying delayed death in *T. gondii* is distinct from that in *Plasmodium*. Delayed death in *T. gondii* is associated with defects in apicoplast biogenesis and formation of parasites that lack apicoplasts during the first lytic cycle (*van Dooren et al., 2009*; *Jacot et al., 2013*; *Lévêque et al., 2015*). Though these 'apicoplast-minus' parasites are viable in the first lytic cycle, they cannot replicate in the second lytic cycle. (In contrast, *Plasmodium* parasites treated with translation inhibitors exhibit apicoplast biogenesis defects in the second replication/lytic cycle, wherein loss of the apicoplast immediately blocks parasite replication [*Dahl et al., 2006*]). Consistent with an apicoplast biogenesis defect in *T. gondii*,

actinonin showed delayed death growth inhibition (EC$_{50}$ = 14 µM; 95% CI 13–14) with formation of apicoplast-minus parasites during drug treatment (*Figure 2a*; *Figure 2—figure supplement 1a*) (*van Dooren et al., 2009*).

Because actinonin disrupted apicoplast biogenesis in both *T. gondii* and *P. falciparum,* suggesting a common target, we selected actinonin-resistant *T.gondii* at a concentration equal to 3xEC$_{50}$ (*Figure 2b*; *Figure 2—figure supplement 1b*). We determined the whole-genome sequences for eight independently selected clones (*Figure 2—source data 1*). Remarkably, five clones, which showed between 3.5 and 4-fold shift in the actinonin EC$_{50}$, harbored two identical mutations

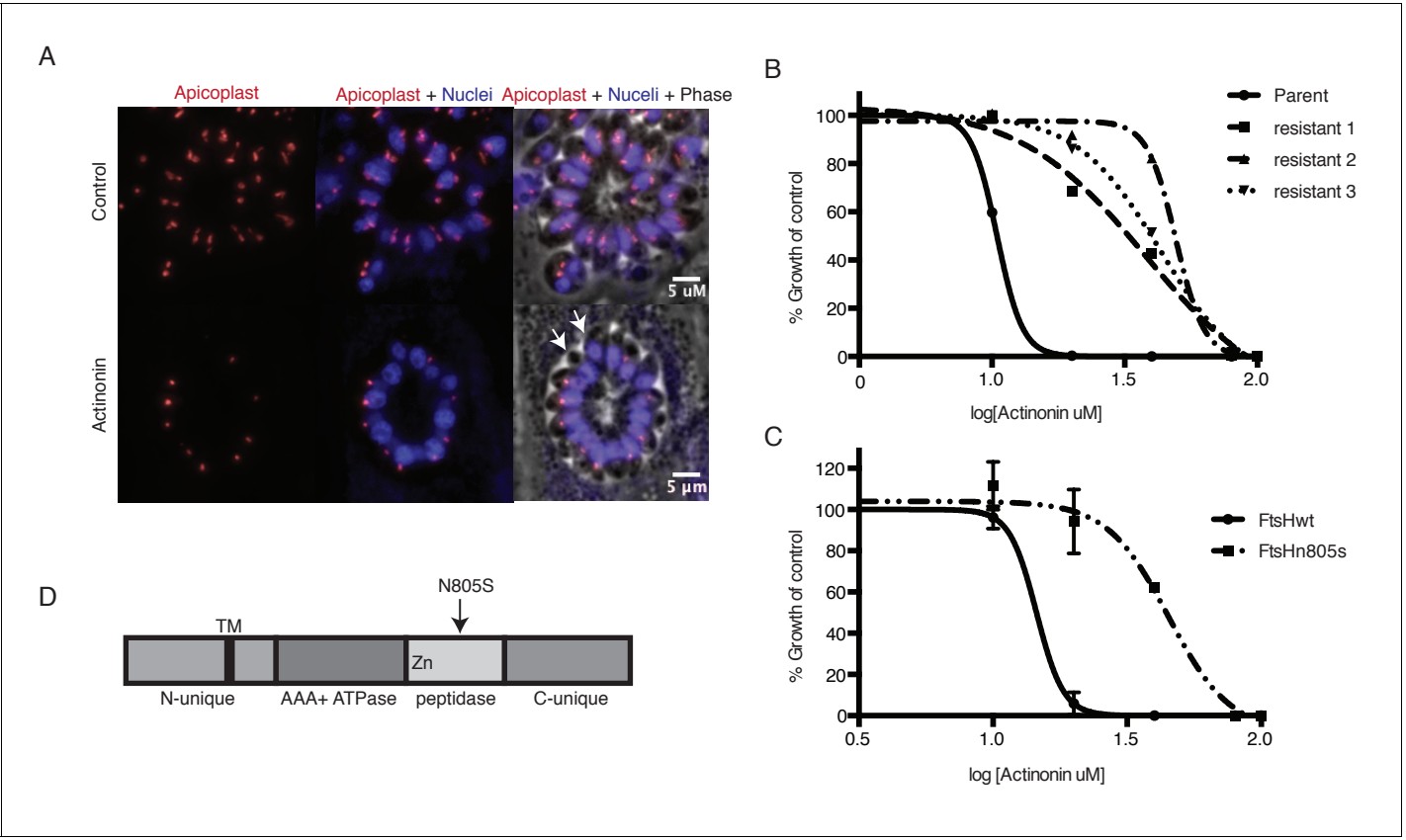

**Figure 2.** A mutation in the protease domain of *ftsH1* is sufficient to confer resistance to actinonin in *T. gondii.* (a) Representative images of the apicoplast of control and actinonin treated parasites 36 hr after infection. The apicoplast is visualized using the *T. gondii* reporter strain RH FNR-RFP in which RFP is targeted to the apicoplast and the nucleus is stained with Hoescht 33342. Each parasite contains one apicoplast, except during cell division when there may be two. White arrows point at examples of *T. gondii* parasites missing an apicoplast. (b) Dose-dependent parasite growth inhibition upon treatment with actinonin for the actinonin-sensitive parent strain (RH) compared with 3 independent clones following selection for actinonin resistance (resistant 1, resistant 2, resistant 3). These three resistant clones are representative of the eight clones submitted for whole genome sequencing. Growth was measured via summed areas of the plaques formed during plaque assays and normalized to untreated controls. Error bars represent the SEM of two biological replicates. (c) Dose-dependent parasite growth inhibition upon treatment with actinonin for *ftsH1WT* compared with *ftsH1(N805S)* parasites in RH Δku80 strain. Data was measured and analyzed as in 2b. (d) Schematic of TgFtsH1. This protein contains a *N*-unique region containing a putative transmembrane domain, an AAA ATPase domain used for unfolding proteins, a peptidase domain with a zinc co-factor in the catalytic site, and a *C*-unique region. The resistance-conferring variant FtsH(N805S) is found in the peptidase domain near the catalytic site.
DOI: https://doi.org/10.7554/eLife.29865.008

The following source data and figure supplement are available for figure 2:

**Source data 1.** Mutations discovered from whole-genome sequencing of actinonin-resistant *T. gondii.*
DOI: https://doi.org/10.7554/eLife.29865.010

**Source data 2.** Numerical data for *Figure 2* and *Figure 2—figure supplement 2* .
DOI: https://doi.org/10.7554/eLife.29865.011

**Figure supplement 1.** Actinonin treatment causes 'delayed death' in *T. gondii* associated with apicoplast loss.
DOI: https://doi.org/10.7554/eLife.29865.009

(*Figure 2—source data 1*). The first mutation encoded a N805S variant in the metalloprotease domain of the membrane-bound AAA +protease *TgFtsH1* (TGGT1_259260) (*Figure 2d*; *Figure 2— source data 1*). The second was found in a RING finger domain-containing protein of unknown function (TGGT1_232160), leading to a G404R mutation outside of the catalytic domain (*Figure 2— source data 1*). Though both candidates may be involved in the mechanism of resistance to actinonin, *Tg*FtsH1 was more compelling as the actinonin target for three reasons. First, *Tg*FtsH1 localizes to the apicoplast (*Karnataki et al., 2007*). Second, actinonin is a peptide mimetic containing a metal-binding hydroxamic acid, a class of molecules that typically binds metalloproteases in their active site (*Figure 1a*) (*Chen et al., 2000*; *Ganji et al., 2015*). Third, the N805S variant of *Tg*FtsH1 is within the metalloprotease domain (*Figure 2d*), raising the possibility that actinonin binding to the *Tg*FtsH1 metalloprotease active site may be prevented by this variant. Indeed, replacement of the endogenous *TgFtsH1* locus with the allele encoding *TgFtsH1(N805S)* caused a 3.5-fold shift in the actinonin $EC_{50}$, fully accounting for the resistance observed in the actinonin-resistant clones (*Figure 2c*). Notably growth inhibition at this higher actinonin concentration did not result in delayed death, suggesting that actinonin no longer bound FtsH1 and growth inhibition was the result of secondary targets (*Figure 2—figure supplement 1b*). In contrast, no increase in the actinonin $EC_{50}$ was observed when the endogenous locus was replaced with the WT allele (*Figure 2c*). Taken together, the known metalloprotease binding of actinonin, the predicted metalloprotease activity of TgFtsH1, and the validated actinonin-resistant mutation in *TgFtsH1* support FtsH1 as the target of actinonin in *T. gondii*, providing a strong candidate for validation in *P. falciparum*.

## C-terminal cleavage of the *P. falciparum* FtsH1 homolog is dependent on the apicoplast

The *P. falciparum* genome contains three FtsH homologs. One of these, *Pf*FtsH1 (Pf3D7_1239700) is most closely related to *Tg*FtsH1 by phylogenetic analysis (*Tanveer et al., 2013*). Unlike *Tg*FtsH1, *Pf*FtsH1 was previously reported to localize to the mitochondrion, not the apicoplast (*Tanveer et al., 2013*). However the same study reported that, like *Tg*FtsH1, *Pf*FtsH1 undergoes processing. The mitochondrial localization was most clearly demonstrated for the *C*-terminally tagged fragment, but the localization of the *N*-terminal fragment containing the ATPase and protease domains was unclear. Proteins required for apicoplast biogenesis may localize to various cellular compartments outside of the apicoplast. For example, both the dynamin DrpA and Atg8 required for apicoplast biogenesis are cytoplasmic proteins, whereas apicoplast protein trafficking machinery is likely to exist in the ER (*van Dooren et al., 2009*; *Lévêque et al., 2015*). Close association of the mitochondrion and apicoplast has also been observed and may be important for organelle biogenesis (*van Dooren et al., 2005*; *Stanway et al., 2011*).

To clarify its localization and function, we constructed a *P. falciparum* strain in which the endogenous *PfFtsH1* locus was modified with a *C*-terminal FLAG epitope and 3' UTR tetR-DOZI-binding aptamer sequences for regulated expression (*Ganesan et al., 2016*). We first sought to localize *Pf*FtsH1 with the *C*-terminal FLAG epitope. However only the FLAG-tagged Cas9 expressed in this strain was detected with no visible band corresponding to the size of *Pf*FtsH1, consistent with the previous report that the *C*-terminus of the protein is cleaved (*Figure 3—figure supplement 1a*). *Tg*FtsH1 was proposed to be *C*-terminally cleaved in the apicoplast (*Karnataki et al., 2009*). We hypothesized that if cleavage of *Pf*FtsH1 is a result of apicoplast localization, then IPP-rescued parasites lacking an apicoplast would retain an intact *Pf*FtsH1-FLAG. Indeed, *Pf*FtsH1-FLAG was detected in these apicoplast-minus parasites, indicating that *C*-terminal cleavage of *Pf*FtsH1 is dependent on the apicoplast (*Figure 3—figure supplement 1a*). This evidence strongly suggests that *Pf*FtsH1 traffics to the apicoplast, as has been shown for *Tg*FtsH1.

## Knockdown of *Pf*FtsH1 disrupts apicoplast biogenesis

Using the endogenously tagged and regulated *Pf*FtsH1 strain, we determined whether *Pf*FtsH1 is essential for apicoplast biogenesis. As expected, *Pf*FtsH1 expression was down-regulated when the tetR-DOZI repressor bound the aptamer sequences, and restored when anhydrotetracycline, which disrupts this interaction, was added (*Figure 3a*; *Figure 3—figure supplement 1b*) (*Ganesan et al., 2016*). Upon downregulation of *Pf*FtsH1, we observed a nearly 4-fold decrease in parasitemia after 3 replication cycles, compared to parasites expressing normal levels of *Pf*FtsH1 (*Figure 3b*; *Figure 3—*

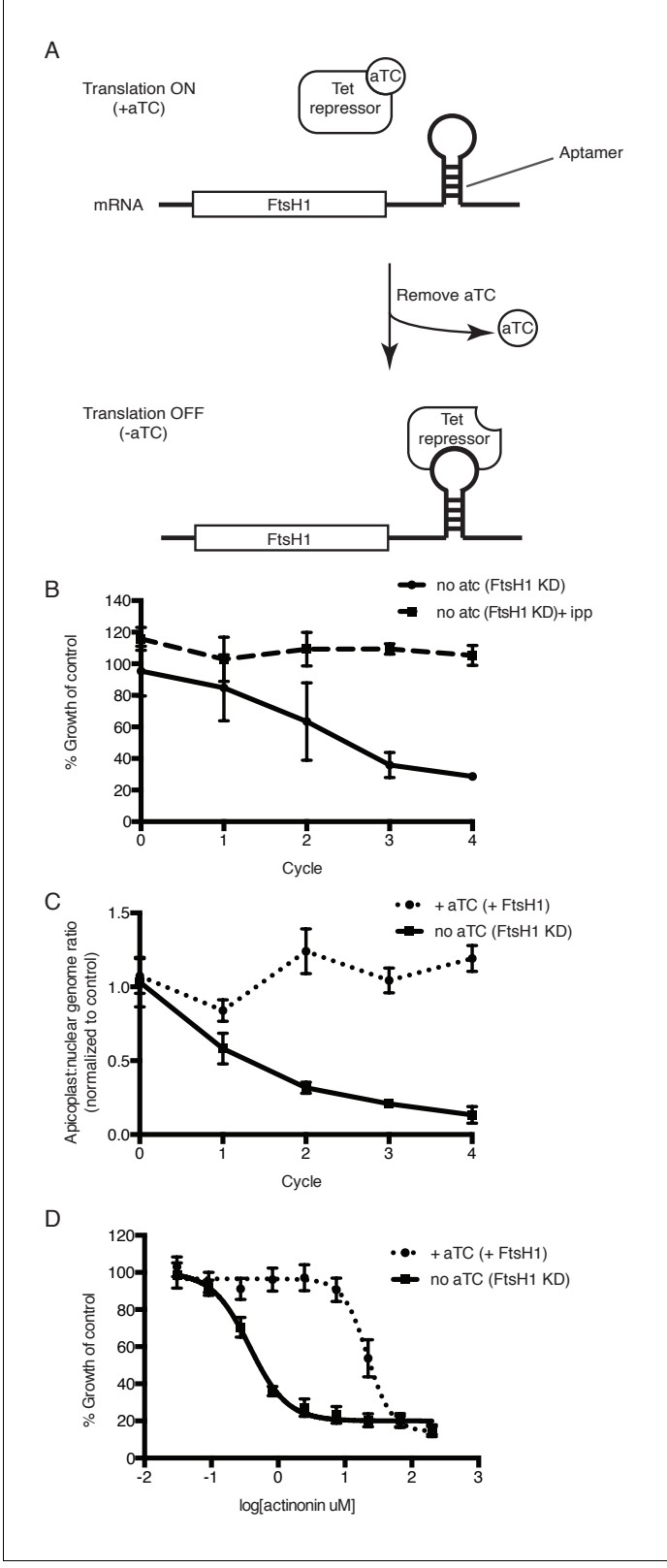

**Figure 3.** Knockdown of *ftsH1* in *P. falciparum* leads to apicoplast loss and hypersensitivity to actinonin. (a) Schematic of the endogenous knockdown strategy. When aTC is present in the media, the tet-repressor binds aTC and does not bind the 10x-aptamer sequence, which relieves translational repression, allowing PfFtsH1 to be expressed. When aTC is washed out of the media, the tet-repressor binds the 10x-aptamer and prevents
*Figure 3 continued on next page*

*Figure 3 continued*

expression of PfFtsH1. (**b**) Time course of parasite growth without aTC and in the presence or absence of IPP in the media, normalized to the untreated or IPP-rescued parental strain as appropriate. Error bars represent the SEM of two biological replicates. (**c**) Time course of the apicoplast:nuclear genome ratio measured by quantitative PCR (qPCR) using primers for the apicoplast and nuclear genomes during treatment with or without aTC. All samples contained IPP to rescue parasite growth. Genome ratios were normalized to respective parental cultures also grown with IPP. Error bars as in c. (**d**) Dose-dependent parasite growth inhibition by actinonin in the absence or presence of aTC. Error bars as in c.

DOI: https://doi.org/10.7554/eLife.29865.012

The following source data and figure supplements are available for figure 3:

**Source data 1.** Oligonucleotide primers and plasmids used in this study.
DOI: https://doi.org/10.7554/eLife.29865.015
**Source data 2.** Numerical data for *Figure 3* and *Figure 3—figure supplement 3* .
DOI: https://doi.org/10.7554/eLife.29865.016
**Figure supplement 1.** *C*-terminal cleavage of PfFtsH1 is dependent on the presence of the apicoplast and efficiency of *Pf*FtsH1 knockdown can be assessed in parasites missing their apicoplast.
DOI: https://doi.org/10.7554/eLife.29865.013
**Figure supplement 2.** Knockdown of *Pf*FtsH1 specifically disrupts the apicoplast and leads to specific hypersensitivity to actinonin.
DOI: https://doi.org/10.7554/eLife.29865.014

*figure supplement 2a*). This growth defect observed over multiple replication cycles reflected the slower kinetics of genetic regulation (compared to chemical inhibition) and partial knockdown, which has been reported using this regulation system (*Ganesan et al., 2016*; *Spillman et al., 2017*). Significantly, growth of *Pf*FtsH1 knockdown parasites was restored by addition of IPP (*Figure 3b*, *Figure 3—figure supplement 2a*). Finally, we confirmed that loss of *Pf*FtsH1 led to apicoplast loss. Using qPCR, we observed a steady decrease in the apicoplast:nuclear genome ratio upon knockdown of *Pf*FtsH1 under IPP rescue compared to control parasites, consistent with loss of the apicoplast (*Yeh and DeRisi, 2011*) (*Figure 3c*). These results demonstrate that *Pf*FtsH1 is required for apicoplast biogenesis and is the first molecular player in apicoplast biogenesis identified in a phenotypic screen.

## Knockdown of FtsH1 results in specific hypersensitivity to actinonin

Because knockdown of *Pf*FtsH1 phenocopies the apicoplast biogenesis defect of actinonin, we sought to determine whether actinonin functionally inhibits *Pf*FtsH1. Although we identified an actinonin-resistant mutation in *Tg*FtsH1 encoding an N805S variant, this residue is not conserved between the *P. falciparum* and *T. gondii* homologs and does not allow the same resistant mutation to be tested (*Karnataki et al., 2007*). As an alternative, we down-regulated *Pf*FtsH1 expression and observed a 58-fold decrease in the actinonin $EC_{50}$ upon knockdown of *Pf*FtsH1 expression (*Figure 3d*). To ensure that the decrease in *Pf*FtsH1 did not cause confounding effects, growth inhibition was measured over a single replication cycle during which *Pf*FtsH1 downregulation did not significantly affect parasite growth (*Figure 3b*; *Figure 3—figure supplement 2b*). Thus reduced levels of *Pf*FtsH1 caused hypersensitivity to actinonin, analogous to drug-induced haploinsufficiency which is commonly used in yeast to identify drug mechanism-of-action (*Giaever et al., 1999*; *Baetz et al., 2004*). Importantly, *Pf*FtsH1 downregulation did not alter the $EC_{50}$ of fosmidomycin, which blocks apicoplast metabolism, or chloramphenicol, which causes delayed biogenesis defects. These negative controls indicate that the functional interaction of *Pf*FtsH1 with actinonin is specific and is not observed with other inhibitors that disrupt apicoplast metabolism or biogenesis (*Figure 3—figure supplement 2c–d*). This specific and strong chemical-genetic interaction indicates that *Pf*FtsH1 is required for actinonin's mechanism-of-action.

## Actinonin inhibits *Pf*FtsH1 activity in vitro

After our results revealed a specific and biologically relevant functional interaction between actinonin and *Pf*FtsH1 in intact parasites, we sought to determine whether actinonin directly inhibits *Pf*FtsH1 enzymatic activity in vitro. PfFtsH1$_{91-612}$ lacking its transmembrane domain was expressed

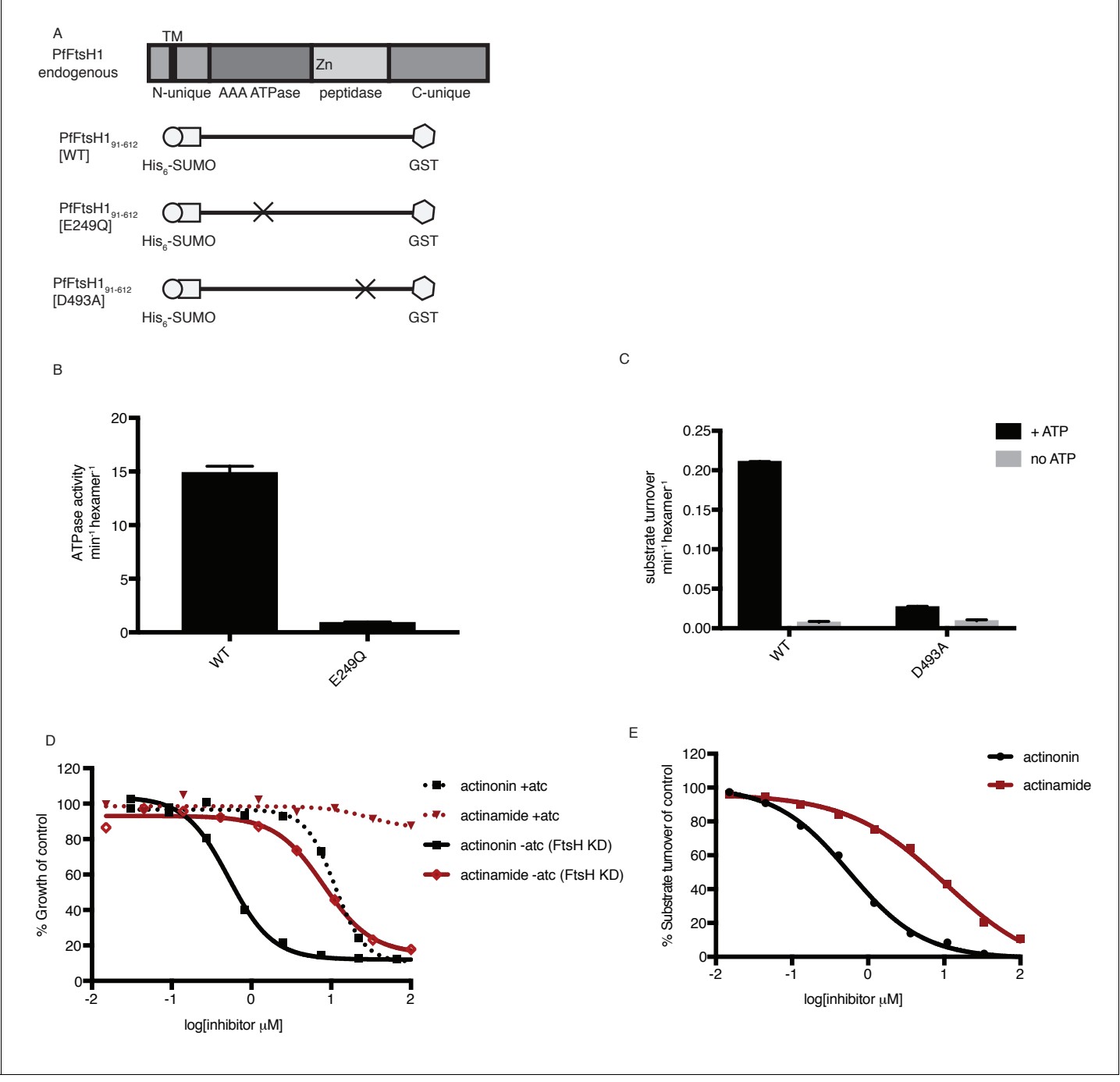

**Figure 4.** Actinonin inhibits *Pf*FtsH1 in vitro. (a) Schematic of *Pf*FtsH1 constructs used for biochemical assays. Amino acids 91–612 of the endogenous protein (PfFtsH1 endogenous), which include the AAA +ATPase and peptidase domains, were placed between His$_6$-SUMO and GST domains to aid in purification and solubility. WT is the parent construct, E249Q is an inactivating mutation in the AAA +ATPase domain, and D493A is an inactivating mutation in the peptidase domain. (b) ATP hydrolysis by *Pf*FtsH1 WT and E249Q measured using a coupled spectrophotometric assay (**NorbyNørby, 1988**). (c) ATP-dependent proteolysis of FITC-labeled casein by *Pf*FtsH1 WT and D493A. (d) Dose-dependent parasite growth inhibition by actinonin (black) or actinamide (red) with and without knockdown of *Pf*FtsH1. Error bars represent the SEM of two biological replicates. (e) Dose-dependent proteolytic inhibition of FITC-labeled casein by *Pf*FtsH1 WT. Error bars represent the SEM of 3 replicates.

DOI: https://doi.org/10.7554/eLife.29865.017

The following source data and figure supplement are available for figure 4:

**Source data 1.** Numerical data for *Figure 4* and *Figure 4—figure supplement 4* .

DOI: https://doi.org/10.7554/eLife.29865.019

*Figure 4 continued on next page*

*Figure 4 continued*

**Figure supplement 1.** Purification of *Pf*FtsH1.

DOI: https://doi.org/10.7554/eLife.29865.018

as a soluble fusion protein and purified using tandem affinity tags (*Figure 4a*; *Figure 4—figure supplement 1*). As expected, the purified enzyme showed ATPase and protease activity (*Figure 4—figure supplement 1*; *Figure 4b–c*). Specific mutations reported to inactivate the ATPase (E249Q) or protease (D493A) domains in other AAA+ proteases (*Hersch et al., 2005*; *Bieniossek et al., 2006*) abolished these respective activities in the *Pf*FtsH1 construct (*Figure 4a–c*) (*Hersch et al., 2005*; *Bieniossek et al., 2006*). Notably, actinonin inhibited *Pf*FtsH1 protease activity with an $IC_{50} \leq 0.6$ μM (*Figure 4d*) (sensitivity limitations of the assay precluded a more accurate number). The metal-binding hydroxamate group of actinonin is known to be very important for inhibition of peptide deformylase (*Chen et al., 2000*). To determine the contribution of the hydroxamate group to the inhibition of *Pf*FtsH1, we tested its inhibition by actinamide, an analog in which the hydroxamate is replaced with an amide. Indeed, actinamide was at least 10-fold less potent against *Pf*FtsH1 ($IC_{50} = 7.3$ μM; *Figure 4e*) (*Lee et al., 2004*). The decreased enzymatic inhibition correlated with a substantial decrease in parasite growth inhibition compared to actinonin, although growth inhibition by actinamide was still dependent on *Pf*FtsH1 levels (*Figure 4d–e*). Hence, the direct inhibition of *Pf*FtsH1 enzymatic activity by actinonin demonstrates that *Pf*FtsH1 is its target in *P. falciparum*.

## Discussion

We identified actinonin as a novel inhibitor of apicoplast biogenesis based on its distinct apicoplast inhibition phenotype compared to known inhibitors. Because the phenotypic screen and the mechanism-of-action elucidation were unbiased, the identification of apicomplexan FtsH1 as actinonin's molecular target was surprising in a number of ways. First, the actinonin target in bacteria and mammalian mitochondria is PDF, and *Pf*PDF was presumed to be the target in *Plasmodium* parasites (*Chen et al., 2000*; *Lee et al., 2004*). However, our functional assays did not support this hypothesis. Identification of antimalarial drug targets based on homology with known drug targets in distantly related organisms can be misleading. For example, the misidentification of fatty acid enzyme *Pf*FabI as the target of triclosan, based on its inhibition of bacterial homologs, led to failed drug development programs when fatty acid biosynthesis was later shown to be dispensable in blood-stage *Plasmodium* (*Vaughan et al., 2009*; *Yu et al., 2008*; *Surolia and Surolia, 2001*). Instead, consistent with actinonin's chemical structure, both bacterial PDF and apicomplexan FtsH1 are metalloproteases. The target in each organism will depend on its biological activity in the cell and relative binding affinity to actinonin.

Second, though *P. falciparum* and *T. gondii* share conserved apicoplast biogenesis pathways, it was uncertain whether *T. gondii*'s tractable culture system and genetics could be leveraged to identify the actinonin target in *P. falciparum*. Fortunately, actinonin caused apicoplast biogenesis defects in both *P. falciparum* and *T. gondii* supporting a common mechanism-of-action. Specifically, actinonin-treated *P. falciparum* formed 'apicoplast-minus parasites' with IPP rescue while *T. gondii* formed them spontaneously (*Yeh and DeRisi, 2011*; *He et al., 2001*). Importantly we show that, unlike in *T. gondii*, disruption of apicoplast biogenesis does not necessarily lead to delayed death in *P. falciparum*. Based on our experience, drug resistance selection in *T. gondii* to aid in elucidating antimalarial mechanism-of-action, where resistant *P. falciparum* cannot be selected, is a powerful approach. The evidence for FtsH1 as the common actinonin target in these two related organisms greatly strengthens the target identification.

Third, we identified an actinonin-resistant mutation in *Tg*FtsH1 but, since *P. falciparum* contains 3 FtsH homologs, it was not initially clear which of them was the *Tg*FtsH1 ortholog. Almost all eukaryotes contain at least two FtsH homologs located in the mitochondrial inner membrane: The i-AAA has a single transmembrane domain with catalytic domains facing the intermembrane space, while the m-AAA has two transmembrane domains and faces the matrix (*Janska et al., 2013*). PF3D7_1119600 is the only homolog predicted to have two transmembrane domains and phylogenetically clusters with human and yeast m-AAA (*Tanveer et al., 2013*). Of the two homologs with a single transmembrane domain, PF3D7_1464900 is more closely related to mitochondrial i-AAA from

human and yeast, leaving *Pf*FtsH1 as the apparent ortholog of *Tg*FtsH1 (*Tanveer et al., 2013*). However this pairing contradicted localization studies assigning *Tg*FtsH1 to the apicoplast and *Pf*FtsH1 to the mitochondrion (*Karnataki et al., 2007*; *Tanveer et al., 2013*). *Tg*FtsH1 was shown to undergo *C*-terminal processing associated with its trafficking to the apicoplast (*Karnataki et al., 2009*). Similarly, we showed that *Pf*FtsH1 undergoes *C*-terminal processing dependent on the presence of the apicoplast. This shared functional phenotype strongly suggests that *Pf*FtsH1 also traffics to the apicoplast. Therefore, the most parsimonious assignment for the three *Plasmodium* FtsH homologs is PF3D7_1119600 and PF3D7_1464900 are mitochondrial and PfFtsH1 is in the apicoplast. Unfortunately, we were unable to generate endogenously-tagged knockdown strains of either PF3D7_1119600 or PF3D7_1464900 to perform localization and functional studies (see Materials and methods).

Finally, there are multiple potential metalloproteases in *P. falciparum* and *T. gondii* that may be inhibited by actinonin. However, our evidence demonstrates that the primary target of actinonin in these parasites, associated with its disruption of apicoplast biogenesis, is FtsH1. Using IPP rescue to gauge apicoplast specificity, we assayed for secondary targets in *P. falciparum* and detected a non-IPP rescuable target at an actinonin concentration 20-fold higher than that for its apicoplast-specific target (*Figure 1—figure supplement 1a*). Similarly, in *T. gondii*, the loss of 'delayed death' growth inhibition (and its associated formation of apicoplast-minus parasites) signified actinonin's inhibition of a secondary target at a concentration 3.5-fold greater than its specific inhibition of *Tg*FtsH1 (*Figure 2—figure supplement 1b*). Moreover, actinonin preferentially targets metalloproteases from MEROPS clan MA composed of structurally homologous proteins with a conserved metal-binding HEXXH motif and catalytic Glu (*Rawlings and Barrett, 2017*). Out of six metalloproteases predicted to localize to the apicoplast, only FtsH and PDF are in clan MA (*Bracchi-Ricard et al., 2001*; *Tanveer et al., 2013*; *van Dooren et al., 2002*; *Mallari et al., 2014*; *Ponpuak et al., 2007*; *Chen et al., 2006*). Curiously, *Pf*FtsH1 is the first FtsH homolog shown to be inhibited by a small molecule.

Overall, we conclude that apicomplexan FtsH1 is the target of actinonin based on (1) an actinonin-resistant N805S variant in *Tg*FtsH1, (2) disruption of apicoplast biogenesis upon *Pf*FtsH1 knockdown, phenocopying actinonin's biogenesis defect, (3) specific actinonin-induced sensitization to *Pf*FtsH1 knockdown, and (4) in vitro inhibition of *Pf*FtsH1 protease activity by actinonin. The unpredicted outcome underscores the power of unbiased screens to uncover novel drug targets in unique but poorly characterized cellular pathways. As the first phenotypic screen for drug targets in essential apicoplast biogenesis pathways, the approach highlighted in our study provides a framework for similar screens in the future.

Our finding also presents an exciting opportunity for antimalarial drug discovery and clearly demonstrates that there are still untapped opportunities to target the apicoplast for antiparasitic therapy. FtsH1 inhibitors will have significant advantages over existing antimalarials that target apicoplast metabolism or gene expression. Whereas metabolic needs vary throughout the parasite lifecycle and even between the same stage of different *Plasmodium* species (*Srivastava et al., 2016*; *Shears et al., 2015*), apicoplast biogenesis is required at every proliferative stage of the parasite lifecycle and is highly conserved among apicomplexan parasites. For example, apicoplast translation inhibitors have broad clinical application as malaria prophylaxis and a partner drug, in combination with faster-acting compounds, for acute malaria, toxoplasmosis, and babesiosis. In fact, the utility of these antibiotics as antiparasitics would be greater if not for their slow activity. Inhibition of FtsH1 retains all the benefits of targeting apicoplast biogenesis with no delay in the onset-of-action. Moreover, our inability to select actinonin-resistant *Plasmodium* contrasts with the ready selection of in vitro resistance against antibiotics and MEP inhibitors and indicates a lower likelihood of clinical resistance to FtsH1 inhibitors (*Wu et al., 2015*; *Dharia et al., 2009*; *Guggisberg et al., 2014*; *Sidhu et al., 2007*). Taken together, FtsH1 inhibitors have potential for rapid onset, multi-stage efficacy against multiple parasitic infections, and minimal clinical resistance.

In addition to its promising biological properties, FtsH1 is a druggable target. There are clear precedents for active-site inhibitors of metalloproteases advancing as clinical candidates in human trials. Clinically tested inhibitors of human matrix metalloproteases (MMP; e.g. marimastat, rebimastat) and bacterial PDF (e.g. GSK1322322, LBM-415, BB83698) were also peptide mimetics with a metal-binding group (*Vandenbroucke and Libert, 2014*; *Sangshetti et al., 2015*). Based on the development of MMP and PDF inhibitors, metalloprotease inhibitors require optimization for

metabolic stability and selectivity. For example, hydroxamate-based inhibitors, such as actinonin, are labile to liver metabolism, which can be addressed by replacement of this metal-binding group with stable bioisosteres (*Skipper et al., 1980*). Similarly, selectivity depends on specific recognition of protein features beyond the active-site metal, which is shared by many metalloproteases. Notably, bacterial PDF inhibitors have shown excellent safety profiles in human clinical trials, demonstrating that metal-chelating peptide mimetics can achieve selectivity for pathogen targets allowing for their use as acute treatment (*Sangshetti et al., 2015*). An alternative strategy is to identify small molecule binding pockets outside of the FtsH1 metalloprotease active site that may affect regulation of its protease activity (*Vandenbroucke and Libert, 2014*). Such small molecule regulation has been demonstrated for the functionally-related bacterial ClpP proteases (*Gersch et al., 2015*). Both the in vitro PfFtsH1 activity assay and the *P. falciparum* FtsH1 knockdown strain reported in our study can be adapted to perform high-throughput screens for optimized FtsH1 inhibitors. Structure-function studies of *Pf*FtsH1 will also aid in design of active site and allosteric inhibitors. Importantly, optimization of *Pf*FtsH1 inhibitors will benefit enormously from a 'piggyback' strategy to access compound libraries, counterscreens for off-target activity against human metalloproteases, and knowledge about drug properties of metalloprotease active-site inhibitors, which have already been established for bacterial PDF inhibitor programs (*Sangshetti et al., 2015*).

Finally, while molecular players involved in apicoplast biogenesis have been identified based on functional conservation and bioinformatics screens for apicoplast proteins (*Sheiner et al., 2011*), apicomplexan FtsH1 is the first novel molecular player in apicoplast biogenesis identified in a phenotypic screen. Acquired by secondary endosymbiosis of an alga, the apicoplast is evolutionarily distinct. FtsH1's role in organelle biogenesis is not conserved in homologs found in mitochondria or primary chloroplasts and likely represents a novel pathway unique to secondary endosymbionts in this parasite lineage (*Vaishnava and Striepen, 2006*; *van Dooren et al., 2009*; *Moore et al., 2008*; *Spork et al., 2009*). FtsH homologs have broad substrate specificity to perform general degradation of misfolded membrane proteins (*Janska et al., 2013*). However they have also been shown to catalyze the proteolysis of native cytosolic proteins under specific conditions critical for cellular regulation (*Bittner et al., 2017*). For example, the essential function of *E. coli* FtsH is its regulated proteolysis of LpxC, the key enzyme in lipopolysaccharide biosynthesis (*Ogura et al., 1999*). In *Bacillus subtilis,* FtsH-dependent degradation of SpoIVFA regulates spore differentiation (*Rudner and Losick, 2002*). Similarly, *Ec*FtsH also regulates the lysis-lysogeny decision by phage lambda by degrading a CII transcription factor (*Shotland et al., 1997*) and regulates levels of intracellular $Mg^{2+}$ by degrading a $Mg^{2+}$ transporter MgtA (*Wang et al., 2017*). Based on these regulatory functions of bacterial FtsH homologs, we propose that apicomplexan FtsH1 regulates the proteolysis of key apicoplast membrane protein(s) during parasite replication. Ongoing experiments to understand the molecular mechanism of FtsH1 are focused on identification of FtsH1 substrates. A candidate substrate is the RING finger domain-containing protein associated with actinonin-resistant *T. gondii.* FtsH1 offers a rare foothold into a novel apicoplast biogenesis pathway evolved from secondary endosymbiosis and will yield deeper insight into the molecular mechanisms of eukaryogenesis.

## Materials and methods

### Chemicals

Fosmidomycin was purchased from Santa Cruz Biotechnology (Santa Cruz, CA) and 10 mM aliquots were prepared in water. Chloramphenicol was purchased from Sigma Aldrich (Saint Louis, MO) and 50 mM aliquots were prepared in 100% ethanol. Actinonin was purchased from Sigma Aldrich and 25 mM aliquots were prepared in 100% ethanol. Anhydrotetracycline was purchased from Sigma Aldrich and 2.5 mM aliquots prepared in 100% ethanol and used at a final concentration of 0.5 uM. Actinamide was a gift from Drs. David Scheinberg and Ouathek Ouerfelli (Memorial Sloan Kettering) and 25 mM aliquots were prepared in 100% ethanol.

Enoxacin, ciprofloxacin, levofloxacin, norfloxacin, novobiocin, coumeramycin, mericitabine, 2'deoxy-2-F-cytidine, gemcitabine, ADEP1a, beta-lactone 4, beta-lactone 7, and rifampin were acquired and solubilized as noted in *Figure 1—source data 1*.

Isopentenyl pyrophosphate (IPP) was purchased from Isoprenoids LC (Tampa, FL) and stored at 2 mg/mL in 70% methanol, 30% 10 mM ammonium hydroxide at −80C. To prevent methanol toxicity,

aliquots of IPP were dried in the speed vacuum centrifuge before adding to cultures. All drugs were stored at −20C and resuspended just prior to use.

## Plasmodium falciparum culture and transfections

*P. falciparum* D10 (MRA-201), and D10 ACP$_L$-GFP (MRA-568) were obtained from MR4. *P. falciparum* NF54$^{attB}$ was a gift from David Fidock (Columbia University). NF54 $^{attB}$ strain constitutively expressing Cas9 and T7 Polymerase, generated previously (*Sidik et al., 2016*), was used in this study. Parasites were maintained in human erythrocytes (2% hematocrit) in RPMI 1640 media supplemented with 0.25% Albumax II (GIBCO Life Technologies, Waltham, MA), 2 g/L sodium bicarbonate, 0.1 mM hypoxanthine, 25 mM HEPES (pH 7.4), 50 μg/L gentamycin, and 0.4% glucose at 37°C, 5% O$_2$, and 5% CO$_2$. All cell lines tested negative for mycoplasma contamination during routine checks.

Parasites were transfected using methods already published (*Ganesan et al., 2016*). Briefly, we used 50 ug of plasmid per 200 uL packed red blood cells (RBCs) adjusted to 50% hematocrit. We used a Bio-Rad Gene Pulser II to preload uninfected RBCs using eight square-wave pulses of 365 V for 1 ms, separated by 100 ms. Preloaded RBCs were resealed for 1 hr at 37C and washed twice in RPMI to remove lysed cells. Schizont stage parasites at 0.5% parasitemia were then allowed to invade half of the preloaded RBCs during two sequential reinvasions. Media was changed daily for the first 12 days and every other day thereafter. Parasites were split 1:1 into fresh blood every 4 days until parasites were visible by Giemsa smear. To select for integration of the pFtsH1 into *P. falciparum* NF54$^{attB-pPfCRISPR}$ parasites, transfected parasites were maintained in media containing 5 nM WR99210 and 0.5 uM anhydrotetracycline (Sigma) and then selected with 2.5 mg/l Blasticidin S (Sigma) beginning 4 days after transfection. Transfected parasites were authenticated using PCR to check for integration (*Figure 3—source data 1*), followed by sequencing of the modified locus, and western blot to visualize tagged proteins at the expected size (*Figure 3—figure supplement 1*).

## Toxoplasma gondii culture and transfection

*T. gondii* RH and *T. gondii* RH Δku80Δhxgprt strains were a gift from Matthew Bogyo (Stanford University) and maintained by passage through confluent monolayers of human foreskin fibroblasts (HFFs) host cells. HFFs were cultured in DMEM (Invitrogen) supplemented with 10% FBS (Fetal Plex Animal Serum from Gemini, West Sacramento, CA), 2 mM L-glutamine (Gemini), and 100 ug penicillin and 100 ug streptomycin per mL (Gibco Life Technologies), maintained at 37 C and 5% CO$_2$. Parasites were harvested for assays by syringe lysis of infected HFF monolayers.

For transfection of *T. gondii* Δku80Δhxgprt, 15 ug of the pTgCRISPR plasmid (*Shen et al., 2014*) was combined with 3 ug of the pFtsH1N805S or pFtsH1WT that had been linearized by NotI digestion. Approximately 10$^7$ parasites were released from host cells using syringe lysis and washed into 400 uL of cytomix containing both plasmids. Parasites were eletroporated (BTX Electro Cell Manipulator 600) with 1.2–1.4 kV, 2.5 kV/resistance, R2 (24 ohm) and then allowed to recover in the cuvette at room temperature for 10 mins before adding to host cells. After 24 hr, media containing 25 ug/mL mycophenolic acid (Sigma) and 50 ug/mL xanthine (Sigma) was added to select for transfectants. After a week, plaques were observed and single clones were isolated using limiting dilution. Transfected parasites were authenticated using PCR to check for integration (*Figure 3—source data 1*), followed by sequencing of the modified locus.

## Toxoplasma gondii genome sequencing and SNP identification

Actinonin resistant and susceptible *T. gondii* were grown on 15 cm dishes containing confluent HFF monolayers until spontaneous lysis of the monolayer was achieved. Released parasites were collected and filtered through 5 micrometer syringe filters (Millipore) before isolating DNA (Qiagen DNAeasy Blood and Tissue).

*T. gondii* genomic DNA isolated from either the parental *T. gondii* RH strain (SRR3666219) or any of its derived mutants (SRR3666219, SRR3666222, SRR3666224, SRR3666792, SRR3666794, SRR3666796, SRR3666798, SRR3666799, SRR3666801) was sequenced in an Illumina NextSeq apparatus using 2 × 150 bp reads at an average sequencing depth of 35x. Sequencing reads were quality trimmed and remnants of sequencing adaptors removed with *trimmomatic* (PMID:24695404). Next, reads were mapped to the reference nuclear assembly of the *T. gondii* GT1 strain (ToxoDB v13.0) and the apicoplast genome assembly from the RH strain (NC_001799) with the program *bowtie2*

(PMID:25621011). Duplicated aligned reads were removed with *picard tools* (http://broadinstitute. github.io/picard) and reads spanning InDels were realigned with GATK (PMID:20644199). Afterwards, allelic variants were called with *samtools mpileup* (PMID:19505943) followed by *bcftools call* with *–p* set to 0.05 (PMID:26826718). Finally, classification of mutations was performed with *snpEff* (PMID:22728672).

## Growth inhibition assays

For *P. falciparum* $EC_{50}$ calculations, growth assays were performed in 96 well plates containing serial dilution of drugs in triplicate. Media was supplemented with 200 uM IPP as indicated. Growth was initiated with ring-stage parasites (synchronized with 5% sorbitol treatment) at 1% parasitemia and 1% hematocrit. To calculate growth, cultures were incubated for 72 hr and growth was then terminated by incubation with 1% formaldehyde (Electron Microscopy Sciences, Hatfield, PA) for 30 min at room temperature. Parasitized cells were stained with 50 nM YOYO-1 (Invitrogen) overnight at room temperature and the parasitemia was determined by flow cytometry (BD Accuri C6 Sampler). Data were analyzed by BD Accuri C6 Sampler software. *P. falciparum* growth assays measuring the actinonin $EC_{50}$ were repeated in the laboratory >10 times for WT parasites and >3 times upon knockdown of FtsH1.

For *T. gondii* $EC_{50}$ calculations, plaque assays were performed in 24 well plates containing confluent HFF monolayers serial dilutions of drugs in duplicate. Approximately 50 parasites were counted using flow cytometry and added to each well. After incubating for 6 days, infected monolayers were washed, fixed with methanol for 10 min, stained with 2% crystal violet (Sigma) for 30 min, and then washed again. Plaques were visualized as non-stained areas. The area of each plaque in a given well was measured and summed using ImageJ as a proxy for growth and normalized to the vehicle only control. *T. gondii* plaque assays measuring the actinonin $EC_{50}$ were repeated in the laboratory >2 times for resistant parasites that arose from actinonin selection, >10 times for WT parasites, and >3 times for TgFtsH(N805S).

For measuring the growth inhibition of *P. falciparum* during the time course, 10 uM actinonin, 10 uM fosmidomycin, 30 uM chloramphenicol, 200 uM IPP, and 0.5 uM of anhydrotetracycline was used as necessary. For comparison of growth between different treatment conditions, cultures were carried simultaneously and handled identically with respect to media changes and addition of blood cells. Daily samples were collected and fixed with 1% formaldehyde for 30 min at RT. At the end of the time course, all samples were stained with 50 nM YOYO-1 and parasitemia was measured using flow cytometry. All growth curves were plotted using GraphPad Prism. *P. falciparum* time course experiments were repeated in the laboratory >2 times for WT parasites and >2 times upon knockdown of FtsH1.

For measuring the growth inhibition of *T. gondii* during the time course, 6-well plates were set up with no drug, 40 uM actinonin, 25 nM clindamycin, and 4 uM pyrimethamine. *T. gondii* was added at a MOI = 3. Every 12 hr, parasites were released from HFFs using syringe lysis and counted using flow cytometry (BD Accuri C6 Sampler). After 36 hr, spontaneous lysis of the monolayer was observed and parasites were counted using flow cytometry and then added back to fresh monolayers at MOI = 3 in the absence of drug and parasites were counted every 12 hr as before. *T. gondii* time course experiments were repeated in the laboratory >2 times.

For measuring growth inhibition using tandem-tomato expression as a proxy for growth, tandem-tomato *T. gondii* were seeded onto a black, clear bottom 96 well plate (Costar 3603) at 2000 parasites per well. Parasites were treated at a range of drug concentrations and fluorescence was measured daily in a plate reader (BioTek, Synergy) for 5 days using the bottom read function.

For co-treatments of *P. falciparum* with actinonin and chloramphenicol, 96 well plates containing ring stage parasites at 1% parasitemia and 1% hematocrit were treated with serial dilutions of both actinonin and chloramphenicol alone and in combination. To determine the effect on growth after one lytic cycle, parasites were fixed at 72 hr and parasitemia was measured by flow cytometry as above. To determine the effect on growth after two lytic cycles, 75% of the media was exchanged at 72 hr and plates were incubated for an additional 48 hr following fixation and flow cytometry as above. Media was supplemented with 200 uM IPP as a separate control to insure specificity of the drug at the concentrations used. Co-treatments with actinonin and chloramphenicol were performed a single time in the laboratory using three technical replicates.

## Quantitative real-time PCR

Parasites from 1 mL of *P. falciparum* culture at ring stage were isolated by saponin lysis followed by two washes with PBS. Since the apicoplast genome is replicate during late stages of intraerythrocytic growth, ring stage parasites were used for gDNA isolation each time to insure that a ploidy change did not confound the qPCR results. DNA was purified using DNAeasy Blood and Tissue (Qiagen, Germany). Primers were designed to target genes found on the apicoplast or nuclear genome: tufA (apicoplast) 5'-GATATTGATTCAGCTCCAGAAGAAA-3' and CHT1 (nuclear) 5'-TG TTTCCTTCAACCCCTTTT-3'/5'-TGTTTCCTTCAACCCCTTTT-3'. Reactions contained template DNA, 0.15 uM of each primer, and 1x SYBR Green I Master mix (Roche). qPCR reactions were performed at 56C primer annealing and 65C template extension for 35 cycles on a Applied Biosystem 7900HT system. Relative quantification of target genes was determined (*Pfaffl, 2001*). For each time point, the apicoplast:nuclear genome ratio was calculated relative to the appropriate control collected at the same time. The apicoplast:nuclear genome ratio was measured by qPCR >5 times for WT parasites treated with actinonin and >2 times upon knockdown of FtsH1.

## Fluorescence microscopy

*P. falciparum* D10 ACP(L)-GFP parasites diluted to 0.05% hematocrit were settled on a Lab-Tek II Chambered Coverglass (Thermo Fisher) and incubated in 2 ug/mL Hoescht 33342 stain for 15 min at 37C. Widefield epifluorescence live cell images were acquired with an Olympus IX70 microscope. The microscope was outfitted with a Deltavision Core system (Applied Precision, GE Healthcare, Sunnyvale, CA) using an Olympus x60 1.4NA Plan Apo Lens, a Sedat Quad filter set (Semrock, Rochester, NY) and a CoolSnap HQ CCD Camera (Photometrics, Tucson, AZ). The microscope was controlled and images were deconvolved via softWoRx 4.1.0 software. ImageJ software was used to analyze resulting images.

Live microscopy of *T. gondii* RH FNR-RFP parasites (*Striepen et al., 2000*) was performing using Lab-Tek II Chambered Coverglasses (Thermo) containing confluent HFF monolayers. Parasites were added at an MOI = 1. After 36 hr of incubation, parasites were incubated with 2 ug/mL of Hoescht for 15 min. Widefield epifluorescence live cell images were acquired with a Nikon Eclipse Ti inverted fluorescence microscope with a NA 1.40 oil-immersion objective (Nikon Instruments, Japan) and controlled using MicroManager v1.4. An iXon3 888 EMCCD camera (Andor) was used for fluorescence imaging and an a Zyla 5.5 sCMOS camera (Andor, UK) was used for phase contrast imaging. ImageJ software was used to analyze the resulting images. ACP-GFP parasites treated with actinonin were imaged >4 different times in the laboratory.

## Immunoblot

Parasites from 9 mL of *P. falciparum* culture were isolated by saponin lysis, washed with PBS and resuspended in 1 x NuPAGE LDS sample buffer (Invitrogen). Proteins were separated by electrophoresis on 4–12% Bis-Tris gel (Invitrogen) and transferred to a nitrocellulose membrane. After blocking, membranes were probed with 1:2000 monoclonal mouse anti-FLAG M2 (Sigma) and 1:10,000 IRDye 680RD goat anti-mouse IgG (LiCor Bioscience, Lincoln, NE) for anti-FtsH1 immunoblots. For anti-PDF immunoblots, membranes were probed with 1:2000 rabbit monoclonal anti-MYC (Cell Signaling Technology 2278S, Danvers, MA), followed by 1:20,000 rabbit polyclonal anti-PfAldolase (Abcam ab207494, UK) and 1:10,000 donkey anti-rabbit 800 (LiCor Biosciences). Fluorescence antibody-bound proteins were detected with Odyssey Imager (LiCor Biosciences). When antibodies of the same species were used, membranes were probed and imaged sequentially. Immunoblots of FtsH-FLAG and PFD-myc were repeated in the laboratory >2 times.

## *Toxoplasma gondii* resistance selection

Approximately $2 \times 10^6$ *T. gondii* RH parasites were added to T25s containing a confluent HFF monolayer and allowed to grow for 24 hr. To mutagenize, between 500 uM – 2 mM N-ethyl N-nitrosourea (ENU) diluted in DMSO was added to flasks and incubated for 2 hr at 37 C. Cultures were then washed twice with 10mLs of cold PBS and then released from host cells using syringe lysis. A quarter of the resulting parasites were passaged to T25s containing a fresh monolayer of HFFs. After two passages, parasites were treated with 40 uM actinonin (the minimum inhibitory concentration of sensitive *T. gondii*). After one passage under actinonin selection, a severe bottleneck was observed.

Plaques of resistant parasites could be observed after one week of constant actinonin pressure with periodic media changes. Finally, single clones were isolated using limiting dilution. Actinonin resistance selection was repeated in the laboratory twice.

### *Plasmodium falciparum* construct generation

The primers used for generating different fragments are listed in the *Figure 3—source data 1*. A construct for inducing PfPDF (PF3D7_0907900) expression in *P. falciparum* was generated from the parental pMG96 plasmid. PDF gene was amplified from pUC57-Amp plasmid with the codon optimized PDF gene (PF3D7_0907900) using primers SMG374 and SMG375. The amplicon was cloned in PfCAM base plasmid (pMG96), which contains single aptamer at 5'UTR and 10x aptamer at 3'UTR. The restriction sites MScI and BstEII were used for cloning thus encoding PDF containing c-myc tag at the c-terminal end. The transfection was carried out as discussed previously (*Nkrumah et al., 2006*). A construct for regulating expression of FtsH1 (PF3D7_1239700) in *P. falciparum* (pFtsH1) was generated from the parental pSN054, a modified pJazz linear plasmid (*Figure 3—source data 1*). The left homology region was amplified from parasite genomic DNA using primers SMG476 and SMG 477 and was cloned using FseI and AsiSI restriction sites. FtsH1 protein coding nucleotides 2348–2643 were recoded using gene block (IDT) to remove the PAM site. The right homology region was amplified from parasite genomic DNA using primers SMG501 and SMG502. These fragments were cloned using the I-SceI restriction site. Targeting guide RNA was generated by klenow reaction using primers SMG514 and SMG515 and was inserted using the AflII site. All sequences were ligated into the parent plasmid using Gibson assembly. Constructs for regulating PF3D7_1464900 were generated similarly (SMG505 and SMG506 for the right homology region; SMG495 and SMG496 for the left homology region; SMG518 and SMG519 for the gRNA klenow reaction). While this construct could be generated, no parasites emerged after two transfections. Constructs for regulating PF3D7_1119600 using primers SMG503 and SMG504 for the right homology region and SMG481 and SMG507 for the left homology region were unable to be generated because of unsuccessful PCR.

### *Toxoplasma gondii* construct generation

A construct for knocking in the FtsH1$_{N805S}$ allele into the endogenous loci of FtsH1 (pFtsHN805S) was generated from the parental pTKO2 vector. Briefly, a ~800 bp sequence upstream of the TGGT1_259260 start codon was amplified as the left homology region (using primers KAJ1 and KAJ2). The FtsH$_{N805S}$ sequence was then amplified from actinonin resistant cDNA (using primers KAJ3 and KAJ4). The HXGPRT resistance cassette was amplified off of the pTKO2 cassette (using primers KAJ5 and KAJ6). A ~800 bp sequence downstream of the TGGT1_259260 stop codon was amplified as the right homology region (using primers KAJ7 and KAJ8). To insert the right homology region, the pTKO2 plasmid was cut with HindIII and HpaI and the 800 bp sequence was inserted using infusion (Clontech, Mountain View, CA). To insert the left homology region and the FtsH1$_{N805S}$ allele, the pTKO2 plasmid containing the downstream homology region was cut with NotI and EcoRI and the two PCR products were inserted also using infusion (Clontech). The resulting colonies were tested using a diagnostic HpaI digest and correct clones were subjected to Sanger sequencing of the inserts. To revert the pFtsHN805S construct to pFtsHWT, we used Q5 mutagenesis (New England Biosciences, Ipswich, MA) and primers KAJ9 and KAJ10.

To increase the transfection efficiency and specificity, CRISPR-Cas9 was used to insert a double stranded break at the site of insertion (the endogenous TGGT1_259260 allele). Briefly, pSAG1::Cas9-U6::sgUPRT (*Shen et al., 2014*) was modified to contain a guide sequence specific to a FtsH1 intron using the Q5 mutagenesis kit (NEB) and primers KAJ11 and KAJ12. Sanger sequencing of the guide was used to verify the resulting plasmid.

### Recombinant *Pf*FtsH1 expression and purification

His$_6$-SUMO-*Pf*FtsH1$_{91-612}$-GST was cloned into a pET22-based vector by Gibson assembly (*Gibson et al., 2009*). Point mutations in this gene were constructed by site-directed mutagenesis. Liquid cultures of T7 Express cells (NEB) harboring these plasmids were grown to log phase, cooled to 20°C, and induced with 0.5 mM IPTG plus 0.1% benzyl alcohol (Sigma). Cultures were incubated for an additional 3 hr at 20°C before harvesting and freezing. Cell pellets were resuspended in lysis

buffer (50 mM HEPES, pH 7.5, 200 mM NaCl, 10% glycerol, 2 mM β-mercaptoethanol [βME], 20 mM imidazole), mixed with lysozyme (1 mg/mL) and Benzonase (Sigma), and lysed by sonication. Cleared lysates were incubated with Ni-NTA resin (G-Biosciences, Saint Louis, MO) for 1 hr at 4°C. After extensive washing with lysis buffer, bound proteins were eluted from the resin with lysis buffer plus 300 mM imidazole. The eluent was applied to a GSTrap column (GE Healthcare) and washed with storage buffer (50 mM HEPES, pH 7.5, 200 mM NaCl, 10% glycerol, 2 mM βME) before eluting with storage buffer plus 10 mM reduced glutathione. The eluent was dialyzed overnight against storage buffer, concentrated, snap-frozen, and stored in aliquots at –80°C.

### PfFtsH1 enzymatic assays

Rates of ATP hydrolysis by FtsH were measured using a coupled spectrophotometric assay (*Norby-Nørby, 1988*) in PD buffer (25 mM HEPES, pH 7.5, 200 mM NaCl, 5 mM MgSO4, 10 μM ZnSO4, 10% glycerol) with 3% dimethyl sulfoxide (DMSO) at 37°C. Protein degradation rates were measured by incubating FtsH (1 μM) with FITC-labeled casein (4 μM, Sigma, Type III) in PD buffer plus 3% DMSO. Reactions were started by adding ATP (4 mM) with a regeneration system (16 mM creatine phosphate and 75 μg/mL creatine kinase), and degradation was followed by measuring the fluorescence intensity (excitation 485 nm; emission 528 nm) at 37°C. For activity-inhibition assays, reactions were pre-incubated with inhibitor in DMSO for 10 min before adding ATP. Addition of inhibitor provided the 3% DMSO found in the other assays. For determining specific activities, the total concentration of casein was 10 μM (2 μM of FITC-labeled casein plus 8 μM unlabeled casein).

### Statistical analysis

When applicable, data was analyzed using Graph Pad Prism software and expressed as mean values ± standard error of the mean (SEM). Basic experiments were repeated at least twice including both positive and negative controls. Biological replicates were performed on different days or on independent cultures while technical replicates were performed using cells from the same culture. Experiments were not blinded. All new reagents were validated prior to use. All generated strains were checked for integration using PCR, sequencing of the modified locus, western blot, and microscopy when applicable. All qPCR primers were assessed for single amplicon.

### Data availability

Our research resources, including methods, cells, reagents and protocols, are available upon request. Source data for the whole genome sequencing analysis is provided in *Figure 2—source data 1*.

## Acknowledgements

We thank Katrina Hong for assistance in drug assays, Dr. Boris Striepen for providing the *T. gondii* FNR-RFP as well as tandem tomato strains, Drs. David Scheinberg and Ouathek Ouerfelli for providing actinamide, and Dr. Saman Habib for providing *Pf*FtsH1 antibody. This project has been funded with federal funds from the NIAID, NIGMS, and Director's Fund, National Institutes of Health, Department of Health and Human Services under Award Numbers 1K08AI097239 (EY), 1DP5OD012119 (EY), U19AI110819 (HAL), 1DP2OD007124 (JCN), P50 GM098792 (JCN), AI016892 (RTS), F32GM116241 (SBH), and T32GM007276 (KAJ). Funding was also provided by the Burroughs-Wellcome Fund (EY), Bill and Melinda Gates Foundation (OPP1069759; JCN), and the Stanford Bio-X SIGF William and Lynda Steere Fellowship (KAJ).

## Additional information

### Funding

| Funder | Grant reference number | Author |
| --- | --- | --- |
| National Institutes of Health | 1K08AI097239 | Ellen Yeh |
| Burroughs Wellcome Fund | | Ellen Yeh |

| Bill and Melinda Gates Foundation | OPP1069759 | Jacquin C Niles |
|---|---|---|
| Stanford Bio-X SIGF William and Lynda Steere Fellowship | | Katherine Amberg-Johnson |
| National Institutes of Health | 1DP5OD012119 | Ellen Yeh |
| National Institutes of Health | U19AI110819 | Hernan A Lorenzi |
| National Institutes of Health | 1DP2OD007124 | Jacquin C Niles |
| National Institutes of Health | P50 GM098792 | Jacquin C Niles |
| National Institutes of Health | AI016892 | Robert T Sauer |
| National Institutes of Health | F32GM116241 | Sanjay B Hari |
| National Institutes of Health | T32GM007276 | Katherine Amberg-Johnson |

The funders had no role in study design, data collection and interpretation, or the decision to submit the work for publication.

### Author contributions

Katherine Amberg-Johnson, Data curation, Formal analysis, Validation, Investigation, Visualization, Methodology, Writing—original draft, Writing—review and editing; Sanjay B Hari, Data curation, Formal analysis, Investigation, Methodology, Writing—review and editing; Suresh M Ganesan, Methodology, Writing—review and editing; Hernan A Lorenzi, Resources, Data curation, Formal analysis, Methodology, Writing—review and editing; Robert T Sauer, Jacquin C Niles, Resources, Supervision, Writing—review and editing; Ellen Yeh, Conceptualization, Resources, Supervision, Funding acquisition, Investigation, Writing—original draft, Project administration, Writing—review and editing

### Author ORCIDs

Katherine Amberg-Johnson http://orcid.org/0000-0003-2707-2912
Ellen Yeh http://orcid.org/0000-0003-3974-3816

### Decision letter and Author response

Decision letter https://doi.org/10.7554/eLife.29865.021
Author response https://doi.org/10.7554/eLife.29865.022

## Additional files

### Supplementary files

• Transparent reporting form
DOI: https://doi.org/10.7554/eLife.29865.020

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
