## [Decision Letter]

Thank you for submitting your article "A first-in-class inhibitor of apicomplexan FtsH1 disrupts plastid biogenesis in human pathogens" for consideration by *eLife*. Your article has been reviewed by four peer reviewers, one of whom, Jon Clardy, is a member of our Board of Reviewing Editors and the evaluation has been overseen by Gisela Storz as the Senior Editor. One of the other reviewers, Vasant Muralidharan, has agreed to reveal his identity.

The reviewers have discussed the reviews with one another and the Reviewing Editor has drafted this decision to help you prepare a revised submission.

They were uniformly enthusiastic about the suitability of the manuscript for publication in *eLife*. The enthusiasm focused on the ingenuity of the chemical genetic approach to identifying actinonin's target as PfFtsH1. The reviewers also had a number of suggestions for improvement, and the major points are summarized below:

1) The manuscript focuses more on the molecule, actinonin, than the target while the reviewers felt that the target was of greater significance. The emphasis begins with the title as the phrase 'first-in-class' is usually reserved for drugs. Actinonin is not very potent and its anti-malarial activity, but not its target, were known. The emphasis ends with the concluding paragraph that summarizes the steps needed to transform actinonin from an inhibitor, or biological probe, to a drug. Refocusing the manuscript to focus on how the screen led to a new and likely druggable target rather than the discovery of a potential antimalarial compound is strongly recommended.

2) A deeper discussion of the molecular nature of actinonin, including its structure, would be useful for many readers, and it would also link with potential liabilities of the molecule and target. A short discussion of strengths and weaknesses of protease inhibitors would also be informative to many readers.

3) Several potential ambiguities of the analysis should be addressed including: 1) increased actinonin sensitivity to parasites with lowered levels of FtsH1, 2) the fold change in ftsH1 mutants, 3) identifying the mutations other than those if ftsH1 that conferred resistance, and 4) the possibility that a ploidy change could confound the quantitative real-time PCR results.

---

## [Author Response]

*1) The manuscript focuses more on the molecule, actinonin, than the target while the reviewers felt that the target was of greater significance. The emphasis begins with the title as the phrase 'first-in-class' is usually reserved for drugs. Actinonin is not very potent and its anti-malarial activity, but not its target, were known. The emphasis ends with the concluding paragraph that summarizes the steps needed to transform actinonin from an inhibitor, or biological probe, to a drug. Refocusing the manuscript to focus on how the screen led to a new and likely druggable target rather than the discovery of a potential antimalarial compound is strongly recommended.*

We agree with the reviewers that the identification of *Pf*FtsH1 as a new druggable target is more significant than the demonstrated antimalarial activity of actinonin. To this end, we have refocused the title to highlight the inhibition of FtsH1 instead of the inhibitor. We have removed “first-in-class” from all parts of the manuscript including the title and Abstract to draw attention away from actinonin, which is an early hit and not an optimized drug. The penultimate paragraph describing drug development of actinonin derivatives is now rewritten as a drug development strategy for general FtsH1 inhibitors. The manuscript ends with a discussion of the biological mechanism of FtsH1. We added additional new references that discuss the role of proteolysis of FtsH1 substrates in other organisms and describe how these examples have affected our view of the role of apicomplexan FtsH1 in apicoplast biogenesis.

*2) A deeper discussion of the molecular nature of actinonin, including its structure, would be useful for many readers, and it would also link with potential liabilities of the molecule and target. A short discussion of strengths and weaknesses of protease inhibitors would also be informative to many readers.*

We have included the structure of actinonin in Figure 1. We have extended the Discussion to include a brief description of other similar metalloprotease drug targets that made it to clinical trials, such as the bacterial peptide deformylase (PDF) and human matrix metalloproteases (MMP) targeted for cancer treatment. Like actinonin, the inhibitors developed against these targets bound the metalloprotease active site. We discuss some of the lessons learned from the PDF and MMP inhibitor programs, including ways to increase the selectivity and prevent liver metabolism.

*3) Several potential ambiguities of the analysis should be addressed including: 1) increased actinonin sensitivity to parasites with lowered levels of FtsH1, 2) the fold change in ftsH1 mutants, 3) identifying the mutations other than those if ftsH1 that conferred resistance, and 4) the possibility that a ploidy change could confound the quantitative real-time PCR results.*

1) We specify the exact shift in the actinonin EC_50_ (58-fold) upon knockdown of *Pf*FtsH1 in the updated manuscript.

2) We specify the exact shift in the actinonin EC_50_ (3.5 to 4-fold shift) of the *T. gondii* strains selected for actinonin resistance in the updated manuscript. Importantly the FtsH(N805S) variant fully accounts for the observed resistance.

3) We describe all the mutations that arose in multiple independent selections and include a new summary table of these mutations in Table 2—source data 1. We have another candidate resistance gene (RING finger containing protein; TGGT1_232160) for which validation is in progress. Importantly the FtsH(N805S) fully accounts for the observed resistance, suggesting that this second mutation may be a passenger mutation or compensatory. The scope of this work was to identify a new drug target in apicoplast biogenesis, which the current data strongly support. Validation of the RING domain containing protein will take more than 60 days and is beyond the scope of this work. We believe the other SNV, if it has a role, may address the mechanism of the identified target.

4) We specify that all qPCR samples were taken during the ring stage to avoid noise in our analysis caused by changes in ploidy during apicoplast genome replication.